# Structural insights into cGAMP degradation by Ecto-nucleotide pyrophosphatase phosphodiesterase 1

Kazuki Kato[1], Hiroshi Nishimasu[1], Daisuke Oikawa[2], Seiichi Hirano[1], Hisato Hirano[1], Go Kasuya [1], Ryuichiro Ishitani[1], Fuminori Tokunaga[2] & Osamu Nureki [1]

ENPP1 (Ecto-nucleotide pyrophosphatase phosphodiesterase 1), a type II transmembrane glycoprotein, hydrolyzes ATP to produce AMP and diphosphate, thereby inhibiting bone mineralization. A recent study showed that ENPP1 also preferentially hydrolyzes 2'3'-cGAMP (cyclic GMP-AMP) but not its linkage isomer 3'3'-cGAMP, and negatively regulates the cGAS-STING pathway in the innate immune system. Here, we present the high-resolution crystal structures of ENPP1 in complex with 3'3'-cGAMP and the reaction intermediate pA (3',5')pG. The structures revealed that the adenine and guanine bases of the dinucleotides are recognized by nucleotide- and guanine-pockets, respectively. Furthermore, the structures indicate that 2'3'-cGAMP, but not 3'3'-cGAMP, binds to the active site in a conformation suitable for catalysis, thereby explaining the specific degradation of 2'3'-cGAMP by ENPP1. Our findings provide insights into how ENPP1 hydrolyzes both ATP and cGAMP to participate in the two distinct biological processes.

[1] Department of Biological Science, Graduate School of Science, The University of Tokyo, 7-3-1 Hongo, Bunkyo-ku, Tokyo 113-0033, Japan. [2] Department of Pathobiochemistry, Graduate School of Medicine, Osaka City University, 1-4-3 Asahi-machi, Abeno-ku, Osaka 545-8585, Japan. Correspondence and requests for materials should be addressed to H.N. (email: nisimasu@bs.s.u-tokyo.ac.jp) or to O.N. (email: nureki@bs.s.u-tokyo.ac.jp)

n innate immunity, germ-line-encoded pattern recognition receptors (PRRs) detect pathogen-associated molecular patterns (PAMPs) from pathogens or damage-associated molecular patterns from host cells, thereby activating downstream signaling to induce the production of inflammatory cytokines and type-I interferon[1,2]. Cyclic GMP-AMP synthase (cGAS) is the cytosolic PRR that recognizes non-self DNAs derived from invading viruses[3–5] or bacteria[6–8], as well as self DNAs derived from damaged mitochondria[9] or tumor cells[10,11], to produce cyclic GMP-AMP (cGAMP) from ATP and GTP[12]. cGAMP binds to the endoplasmic reticulum–resident membrane protein STING, thereby inducing the phosphorylation of TANK-binding kinase 1 (TBK1) and interferon regulatory factor 3 (IRF3), followed by the production of interferon-β (IFN-β)[13]. cGAMP is composed of adenosine and guanosine, which are linked via two phosphodiester linkages, and exists as multiple isomers, such as 2′3′-cGAMP and 3′3′-cGAMP (Fig. 1). Biochemical and structural studies revealed that cGAS specifically produces 2′3′-cGAMP, cyclic [G(2′,5′)pA(3′,5′)p], which contains the canonical 3′-5′ and non-canonical 2′-5′ phosphodiester linkages[14–17]. 2′3′-cGAMP binds to STING with higher affinity, as compared to 3′3′-cGAMP, and activates the signaling pathway[18].

ENPP1 (Ecto-nucleotide pyrophosphatase phosphodiesterase 1) is a type II transmembrane glycoprotein originally identified as a negative regulator of bone mineralization[19,20]. The extracellular domain of ENPP1 consists of two N-terminal somatomedin B-like domains (SMB1 and SMB2), a catalytic domain and a nuclease-like domain (Fig. 2a). ENPP1 is expressed on the cell surface in mineralizing cells, such as osteoblasts and chondrocytes, and hydrolyzes extracellular ATP to produce AMP and diphosphate, an inhibitor of bone mineralization. ENPP1 is also expressed in lymphoid organs, and ENPP1-produced AMP is metabolized by the ecto-5′-nucleotidase CD73 to the immunosuppressive adenosine[21]. ENPP1 hydrolyzes nucleotide triphosphates (NTPs), such as GTP and CTP, in vitro, while it preferentially hydrolyzes ATP[22]. The crystal structures of ENPP1 in complex with nucleotide monophosphates revealed the mechanism of NTP recognition and hydrolysis by ENPP1. A recent study reported that ENPP1 also hydrolyzes 2′3′-cGAMP, but not 3′3′-cGAMP, and negatively regulates the cGAS-STING-dependent immune activation[23,24]. However, it remains elusive how ENPP1 hydrolyzes both ATP and 2′3′-cGAMP, and discriminates 2′3′-cGAMP from 3′3′-cGAMP.

Here, we report the crystal structures of ENPP1 in complex with 3′3′-cGAMP and the reaction intermediate pA(3′,5′)

pG (pApG). The structures provided mechanistic insights into the specific degradation of 2′3′-cGAMP by ENPP1, and explain how ENPP1 hydrolyzes ATP and cGAMP to participate in bone mineralization and innate immunity.

## Results

**Structural determination.** To address the ENPP1-mediated 2′3′-cGAMP hydrolysis mechanism, we sought to determine the crystal structure of the extracellular domain of mouse ENPP1 in complex with cGAMP. The SMB domains are dispensable for NTP hydrolysis, and are disordered in the previous structures at moderate (~3 Å) resolutions[22], suggesting that the deletion of the flexible SMB domains could improve the resolution. Thus, we prepared the mouse ENPP1 protein lacking the SMB domains (residues 83–169) (hereafter referred to as ENPP1-ΔSMB) (Supplementary Fig. 1a, b). We confirmed that ENPP1-ΔSMB has in vitro hydrolytic activities toward 2′3′-cGAMP and ATP comparable to those of wild-type ENPP1 (Supplementary Fig. 1c, d). To avoid the 2′3′-cGAMP degradation during crystallization, we replaced the catalytic Thr238 residue with alanine to generate the ENPP1-ΔSMB T238A mutant (Supplementary Fig. 1a). We co-crystallized the ENPP1-ΔSMB T238A mutant in the presence of 2′3′-cGAMP or 3′3′-cGAMP, and determined the crystal structures at 1.8 and 1.9 Å resolutions, respectively (Fig. 2b; Table 1). The two structures consist of the catalytic domain (residues 190–578) and the nuclease-like domain (residues 629–902) (Fig. 2a), and are essentially identical to the ENPP1-AMP complex structure (PDB ID 4GTW, rmsd = 0.65 Å for 700 Cα atoms) (Supplementary Fig. 2a–c), showing that the truncation of the SMB domains does not substantially affect the structures of the catalytic and nuclease-like domains, and resulted in the improvement of the resolution. In the two structures, we observed electron densities corresponding to the dinucleotides bound to the active site in the catalytic domain (Fig. 2c, d). In the co-crystal structure with 2′3′-cGAMP, a continuous density was not observed at the 2′-5′ phosphodiester linkage of 2′3′-cGAMP (Fig. 2c), suggesting that 2′3′-cGAMP was predominantly degraded to the pApG linear intermediate during crystallization, although the catalytically inactive ENPP1-ΔSMB T238A mutant was used for crystallization. This may be due to the slight enzymatic activity of the mutant at a high concentration during crystallization. We concluded that this structure represents a post-reaction state of the first hydrolysis, and modeled pApG into the electron density. In contrast, in the co-crystal structure with 3′

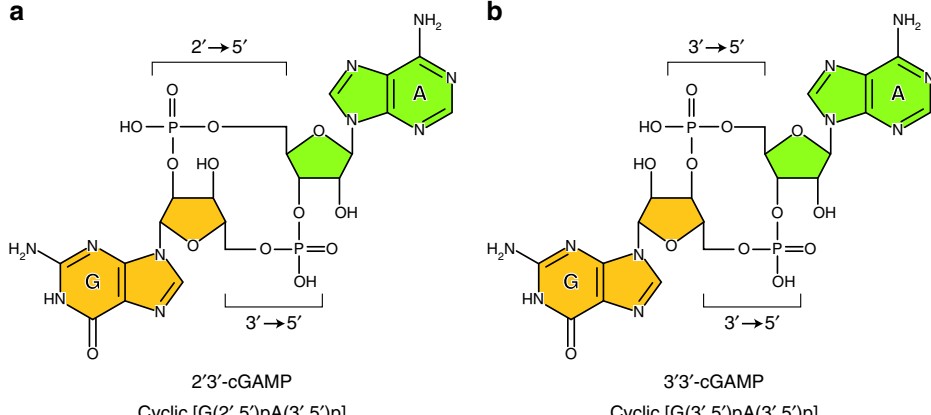

**Fig. 1** cGAMP isomer structures. **a**, **b** Chemical structures of 2′3′-cGAMP, cyclic [G(2′,5′)pA(3′,5′)p] (**a**) and 3′3′-cGAMP, cyclic [G(3′,5′)pA(3′,5′)p] (**b**). 2′3′-cGAMP contains the non-canonical 2′-5′ and the canonical 3′-5′ phosphodiester linkages, while 3′3′-cGAMP contains the two canonical 3′-5′ phosphodiester linkages

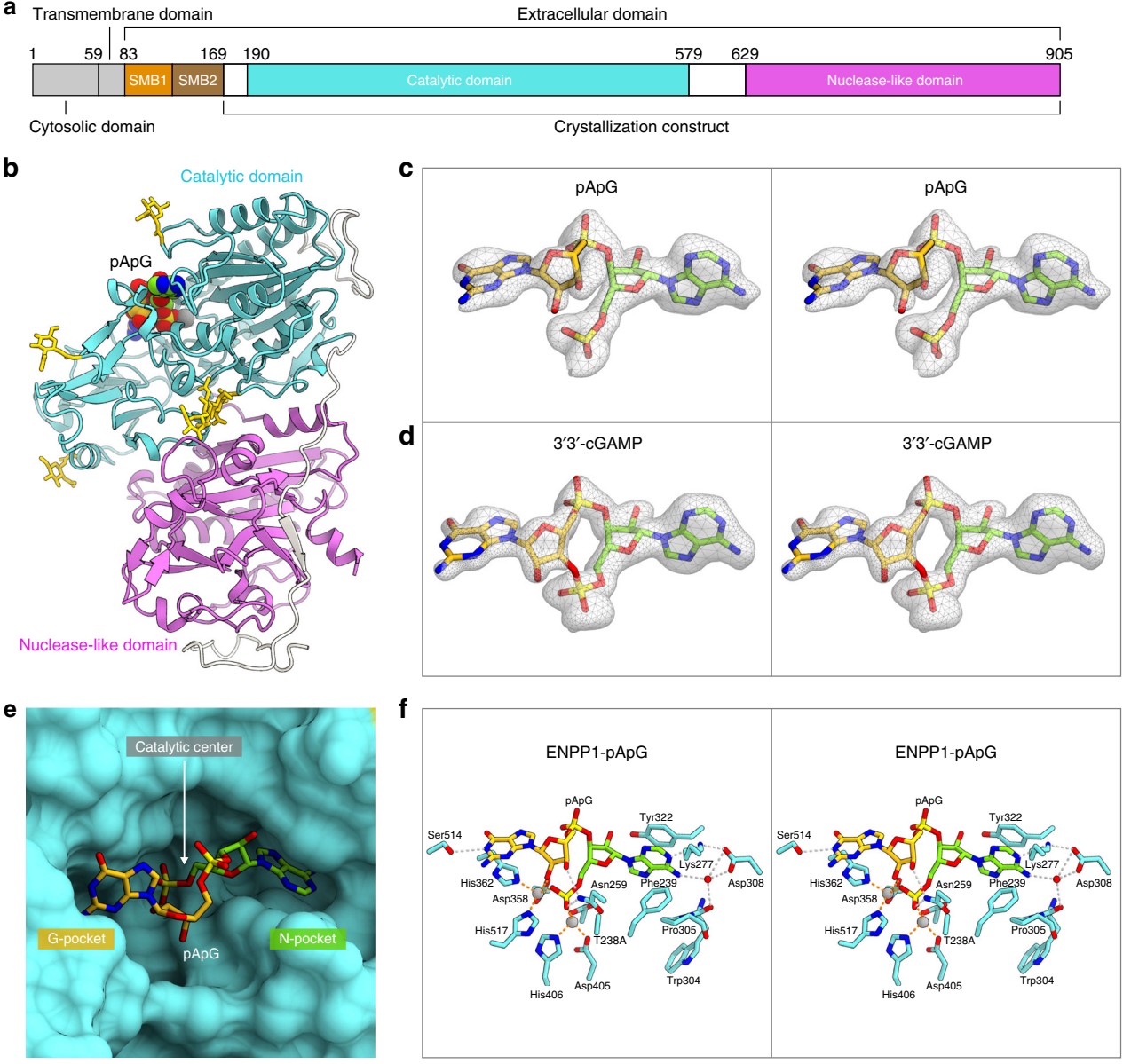

**Fig. 2** Crystal structure of ENPP1 in complex with the dinucleotide. **a** Domain organization of mouse ENPP1. **b** Overall structure of ENPP1 in complex with pApG. The N-linked sugars are shown as yellow sticks. **c**, **d** $mF_O - DF_C$ omit electron density maps for pApG (**c**) and 3′3′-cGAMP (**d**) (contoured at 2.5$\sigma$) (stereo view). **e** Binding of pApG to the ENPP1 active site. **f** Recognition of pApG (stereo view). The zinc ions are shown as gray spheres. Hydrogen bonds are shown as dashed lines

3′-cGAMP, we observed an electron density corresponding to 3′3′-cGAMP (Fig. 2d).

**Dinucleotide recognition.** pApG adopts an open conformation, in which the nucleobases are far apart from each other (Fig. 2e). The adenine and guanine moieties of pApG are accommodated within the nucleotide-binding pocket (N-pocket) and the guanine-binding pocket (G-pocket), respectively. The 5′-phosphate group binds to two zinc ions (Zn1 and Zn2), which are coordinated by Asp358/His362/His517 and Asp200/Asp405/His406, respectively (Fig. 2f). This observation suggests that ENPP1 first hydrolyzes the 2′-5′, rather than the 3′-5′, phosphodiester linkage in 2′3′-cGAMP, consistent with previous data showing that a 2′3′-cGAMP analog with the 2′-5′ non-hydrolyzable phosphothioate diester linkage (2′3′-cGA$^s$MP) is

not degraded by ENPP1[23]. The adenosine moiety of 2′3′-cGAMP is accommodated in the N-pocket formed by Phe239, Lys277, Trp304, Pro305, Asp308, and Tyr322, in a similar manner to that of AMP in the ENPP1-AMP complex structure[22] (Fig. 2f and Supplementary Fig. 2d, e). The adenine base is sandwiched between Phe239 and Tyr322, and forms hydrogen bonds with Lys277 and Asp308 in a base-specific manner. This observation further supports the notion that the 2′-5′ linkage in 2′3′-cGAMP is hydrolyzed by ENPP1 in the first reaction. The guanosine moiety of pApG is accommodated in the G-pocket, formed by His362 on a short helix and a loop (residues 510–517) (Fig. 2f). The 3′-OH group of the guanosine forms a hydrogen bond with Asn259, while the guanine base is stacked with His362, with its N1 atom forming a hydrogen bond with Ser514. The S514L mutation, which would induce steric hindrance with the guanine base, decreased the in vitro hydrolytic activity of ENPP1 toward

**Table 1 Data collection and refinement statistics**

|  | pApG | 3′3′-cGAMP |
|---|---|---|
| *Data collection* |  |  |
| Space group | $P2_1$ | $P2_1$ |
| *Cell dimensions* |  |  |
| $a, b, c$ (Å) | 54.5, 94.1, 74.7, | 53.2, 94.1, 74.3, |
| $\alpha, \beta, \gamma$ (°) | 90.0, 96.9, 90.0 | 90.0, 95.9, 90.0 |
| Resolution (Å) | 47.0–1.80 (1.84–1.80)[a] | 47.0–1.90 (1.94–1.90)[a] |
| $R_{merge}$ | 0.051 (0.662) | 0.078 (0.751) |
| $I/\sigma I$ | 10.1 (1.4) | 9.9 (1.7) |
| Completeness (%) | 99.9 (99.9) | 99.9 (99.9) |
| Redundancy | 3.4 (3.4) | 3.5 (3.5) |
| *Refinement* |  |  |
| Resolution (Å) | 47.0–1.80 | 47.0–1.90 |
| No. reflections | 69,203 | 51,214 |
| $R_{work}/R_{free}$ | 0.183/0.208 | 0.166/0.202 |
| *No. atoms* |  |  |
| Protein | 5817 | 5850 |
| Sugar | 81 | 81 |
| Ligand | 46 | 45 |
| Ion | 9 | 9 |
| Solvent | 346 | 388 |
| *B-factors (Å²)* |  |  |
| Protein | 41.6 | 35.7 |
| Sugar | 48.5 | 44.8 |
| Ligand | 53.4 | 51.2 |
| Ion | 37.5 | 34.8 |
| Solvent | 40.8 | 36.1 |
| *R.m.s. deviations* |  |  |
| Bond lengths (Å) | 0.006 | 0.007 |
| Bond angles (°) | 0.85 | 0.87 |
| *Ramachandran statistics (%)* |  |  |
| Favored | 96.81 | 96.26 |
| Allowed | 3.19 | 3.74 |
| Outlier | 0.00 | 0.00 |

[a]Highest resolution shell is shown in parentheses

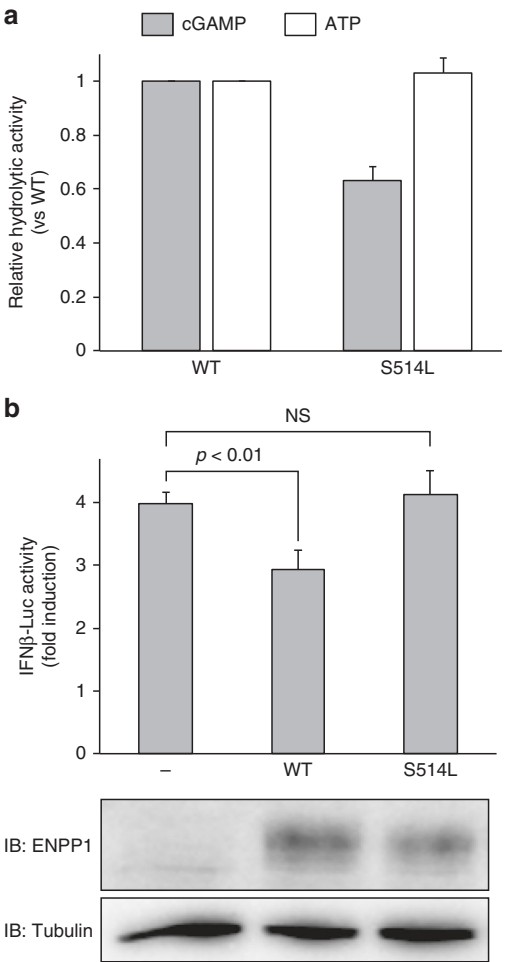

**Fig. 3** Mutational analysis. **a** Hydrolytic activities of wild-type (WT) and the S514L mutant of ENPP1 toward 2′3′-cGAMP or ATP. Data are means ± s.d. ($n = 3$). **b** IFN-β suppression by WT and the S514L mutant of ENPP1. The ENPP1 protein was expressed in THP-1 cells, and the IFN-β induction was then examined by a luciferase assay. Data are means ± s.d. ($n = 3$). One-way ANOVA followed by a post-hoc Tukey HSD test was performed. The cell lysates were immunoblotted with the indicated antibodies. NS, not significant

2′3′-cGAMP, but not ATP (Fig. 3a), indicating the functional importance of the G-pocket for the hydrolysis of 2′3′-cGAMP, but not for ATP. Moreover, the wild-type ENPP1, but not the S514L mutant, suppressed the IFN-β induction, the downstream consequence of STING signaling in human monocytic leukemia THP-1 cell line (Fig. 3b), highlighting the biological importance of the G-pocket for the negative regulation of the STING signaling.

**Linkage specific degradation of 2′3′-cGAMP**. To clarify the preference of ENPP1 for 2′3′-cGAMP over 3′3′-cGAMP, we built a structural model of the ENPP1-2′3′-cGAMP complex based on the ENPP1-pApG complex (Fig. 4a). A structural comparison of the ENPP1-2′3′-cGAMP and ENPP1-3′3′-cGAMP complexes with the ENPP2-orthovanadate ($VO_5$) complex can explain why ENPP1 specifically degrades 2′3′-cGAMP, as compared to 3′3′-cGAMP (Fig. 4a, b). The ENPP family enzymes hydrolyze the phosphodiester linkage in various substrates via a two-step in-line displacement mechanism, in which the conserved threonine residue (Thr238 in ENPP1 and Thr209 in ENPP2) serves as a catalytic nucleophile[25]. The ENPP2-$VO_5$ complex structure represents the phosphoryl transfer intermediate state, in which the Thr209 Oγ atom is aligned in-line with the vanadium atom and the leaving oxygen atom (O1) of the bound $VO_5$ (Fig. 4b). In the ENPP1-2′3′-cGAMP structure, the guanosine moiety of

2′3′-cGAMP adopts the C2′-*endo* conformation, in which its 3′-oxygen atom forms hydrogen bonds with Asn259 and its 2′-oxygen atom is superimposed with the $VO_5$ O1 atom in the ENPP2-$VO_5$ complex (Fig. 4a, c). This observation suggests that the phosphorus atom of the 2′-5′ phosphodiester linkage in 2′3′-cGAMP has a suitable geometry for the in-line attack by the catalytic Thr238. In contrast, in the ENPP1-3′3′-cGAMP structure, the guanosine moiety of 3′3′-cGAMP adopts the C3′-*endo* conformation, which is stabilized by a hydrogen-bonding interaction with Asn259 (Fig. 4a, d). This observation suggests that the phosphorus atom of the 3′-5′ phosphodiester linkage in 3′3′-cGAMP has an unfavorable geometry for the in-line attack. Together, these structural data indicate that 2′3′-cGAMP, but not 3′3′-cGAMP, binds to the ENPP1 active site in a conformation suitable for the in-line attack, thereby explaining the specific degradation of 2′3′-cGAMP by ENPP1.

**Discussion**
Based on the present structures, we propose an ENPP1-mediated 2′3′-cGAMP degradation mechanism, in which the single active

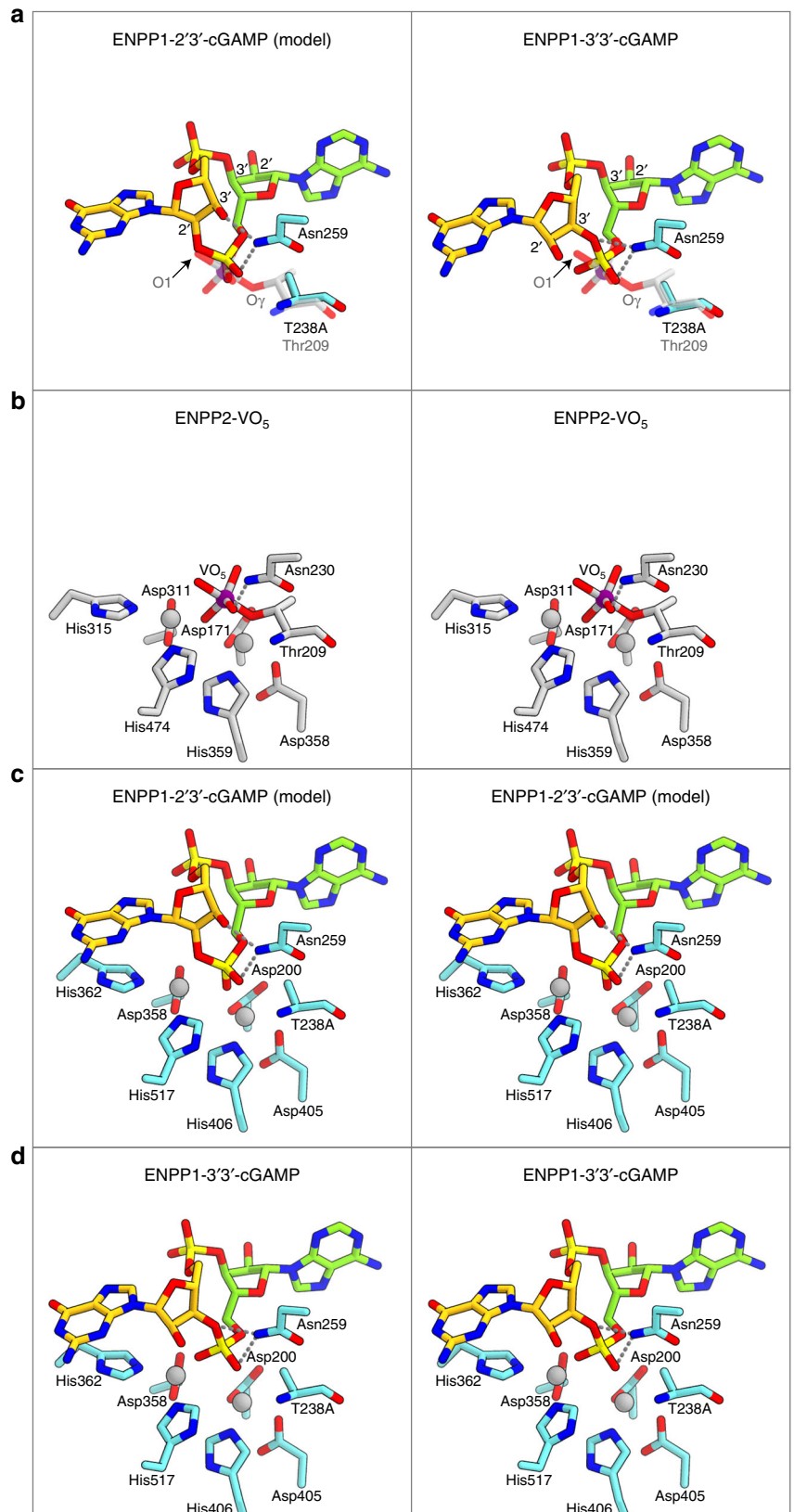

**Fig. 4** Structural comparison between the ENPP1-bound 2′3′-cGAMP and 3′3′-cGAMP. **a** Structural comparison of the ENPP1-2′3′-cGAMP complex (model) (left) and the ENPP1-3′3′-cGAMP complex (right) with the ENPP2-VO$_5$ complex (PDB 5IJS) (semi-transparent). The vanadium atom is shown as a purple sphere. Hydrogen bonds are shown as dashed lines. **b–d** Catalytic centers of the ENPP2-VO$_5$ complex (PDB 5IJS) (**b**), the ENPP1-2′3′-cGAMP complex (model) (**c**), and the ENPP1-2′3′-cGAMP complex (**d**) (stereo views). The zinc and vanadium atoms are shown as gray and purple spheres, respectively. Hydrogen bonds are shown as dashed lines

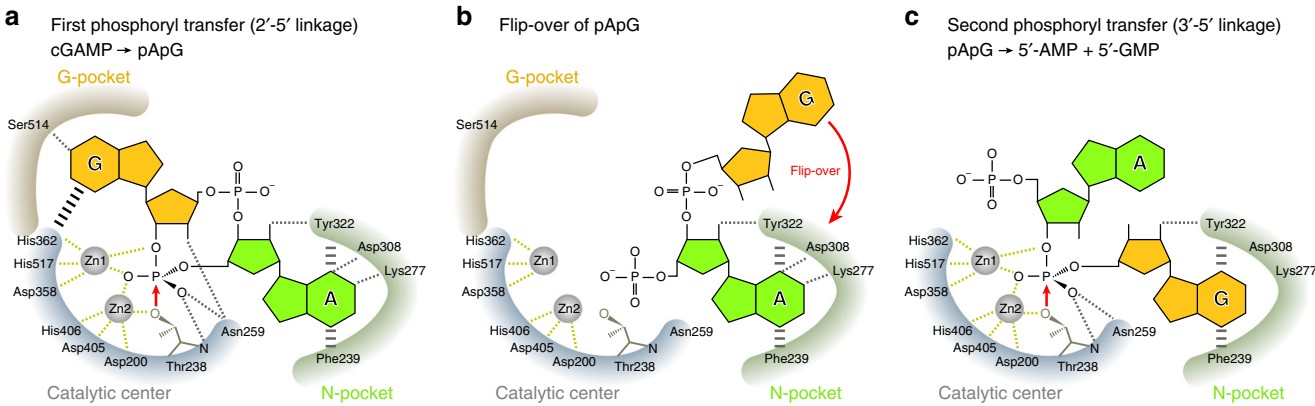

**Fig. 5** Proposed mechanism of the ENPP1-catalyzed 2′3′-cGAMP degradation. **a** ENPP1 accommodates the adenine and guanine bases of 2′3′-cGAMP in the N-pocket and G-pocket, respectively. The Thr238 Oγ atom nucleophilically attacks the phosphorus atom in the 2′-5′ phosphodiester linkage, resulting in the production of the pApG intermediate. **b** The pApG intermediate is flipped over, and the guanine base of pApG is then accommodated in the N-pocket. **c** The 3′-5′ phosphodiester linkage of pApG is hydrolyzed to produce 5′-AMP and 5′-GMP

site of ENPP1 sequentially hydrolyzes the two distinct 2′-5′ and 3′-5′ phosphodiester linkages in 2′3′-cGAMP (Fig. 5). First, ENPP1 recognizes the adenine and guanine bases of 2′3′-cGAMP, in the N-pocket and G-pocket, respectively, and hydrolyzes the 2′-5′ phosphodiester linkage of 2′3′-cGAMP to produce pApG. pApG may then flip-over, as proposed in a previous docking study[24] and the cGAS-catalyzed 2′3′-cGAMP synthesis reaction[17]. The guanine base of pApG is then accommodated in the N-pocket, in a similar manner to that of GMP in the ENPP1-GMP complex structure[22], thereby placing the 3′-5′ phosphodiester linkage of pApG in the catalytic center. ENPP1 then hydrolyzes the 3′-5′ phosphodiester linkage to produce 5′-AMP and 5′-GMP. Given that the pApG intermediate was only faintly detected in a previous study[24] and our in vitro assays (Supplementary Fig. 1d), the pApG intermediate may be rapidly flipped over and degraded to 5′-AMP and 5′-GMP.

ENPP1 preferentially recognizes the adenine base in the N-pocket[22], and efficiently hydrolyzes ATP in the bloodstream to produce diphosphate, which negatively regulates bone mineralization. The present structures indicate that the N-pocket first recognizes the adenine base of 2′3′-cGAMP and then the guanine base of the pApG intermediate, during the sequential cleavages of the phosphodiester linkages of 2′3′-cGAMP, thereby highlighting the key role of the nucleotide preference of the N-pocket for 2′3′-cGAMP degradation. The residues involved in 2′3′-cGAMP recognition are conserved between human and mouse, suggesting a similar mechanism of 2′3′-cGAMP degradation by human ENPP1.

2′3′-cGAMP is produced by cGAS upon viral infection in host cells, whereas 3′3′-cGAMP is a bacterial second messenger recognized by STING as an external PAMP. The present structures explain how ENPP1 degrades the host-produced 2′3′-cGAMP, but not the microbe-derived 3′3′-cGAMP, and specifically suppresses cGAS-dependent immune activation. A recent study identified V-cGAPs as bacterial cGAMP-degrading phosphodiesterases from *Vibrio cholerae*[26]. V-cGAPs specifically hydrolyze 3′3′-cGAMP, rather than 2′3′-cGAMP, to produce pApG. V-cGAPs contain an HD-GYP domain and do not share sequence similarity with ENPP1, suggesting that V-cGAPs and ENPP1 use distinct mechanisms to discriminate between 2′3′-cGAMP and 3′3′-cGAMP. Structural studies of V-cGAPs will provide mechanistic insights into how V-cGAPs specifically hydrolyze 3′3′-cGAMP.

This study revealed the structural basis for the ENPP1-mediated 2′3′-cGAMP degradation. Nonetheless, it still remains elusive how ENPP1 gains access to the cytosolic cGAMP at cellular levels, since ENPP1 is predominantly localized on the plasma membrane and Golgi membrane. One of the possible explanations is that cGAMP may be transported by an unidentified transporter into the extracellular space or Golgi apparatus. Further studies will be needed to elucidate how ENPP1 physiologically regulates the cGAMP-mediated signaling pathway. In summary, our structures provide mechanistic insights into how ENPP1 acts as a bifunctional enzyme degrading ATP and cGAMP, and participates in distinct biological processes, bone mineralization, and innate immunity.

## Methods

**Cloning**. To prepare ENPP1 as a soluble protein, the gene encoding the extracellular domain of mouse ENPP1 (residues 92–905) was fused with an N-terminal signal sequence (residues 1–50) and the N-terminal nine residues of the SMB1 domain (residues 51–59) of mouse ENPP2[27]. The ENPP1 gene was inserted into the modified pcDNA3.1 vector (Invitrogen) containing the C-terminal TEV protease-recognition sequence, followed by the TARGET tag[28]. To prepare the crystallization construct (ENPP1-ΔSMB T238A), the HRV3C protease-recognition sequence was inserted between SMB2 and the catalytic domain (Lys169 and Lys170)[22], and the T238A mutation was introduced by a PCR-based method. The S514L mutant was generated by a PCR-based method, using the plasmid encoding the extracellular domain of ENPP1 as the template. All sequences were verified by DNA sequencing. The primers used in this study are listed in Supplementary Table 1.

**Expression and purification**. The extracellular domain of ENPP1 (T238A) was expressed in HEK293 GnT1⁻ cells[29], which were stably cotransfected with the expression plasmid and the IR/MAR gene (Trans Genic). The culture supernatant containing the secreted ENPP1 T238A protein was mixed with a P20.1-Sepharose resin (Wako) at 4 °C overnight. The resin was washed with buffer containing 20 mM Tris-HCl, pH 7.5, and 150 mM NaCl, and then the protein was eluted with the same buffer containing the TARGET tag peptide (Wako) (0.2 mg/ml). The eluted protein was mixed with the His-tagged HRV3C and TEV proteases, and dialyzed against buffer (20 mM Tris-HCl, pH 8.0, 150 mM NaCl and 20 mM imidazole) at 20 °C overnight. The mixture was passed through a Ni-NTA column (Qiagen) to remove the His-tagged proteases, and then the flow-through fraction was mixed with Endo H glycosidase (NEB) at 20 °C overnight. The ENPP1-ΔSMB T238A protein was further purified using Mono Q (GE Healthcare) and HiLoad Superdex200 16/60 columns (GE Healthcare). For in vitro nucleotide-hydrolyzing assays, the ENPP1 proteins (wild-type and the S514L mutant of the ENPP1 extracellular domain) were expressed in HEK293T cells, and purified using the P20.1-Sepharose column. To prepare ENPP1-ΔSMB, the purified ENPP1 protein was treated with HRV3C protease, and then passed through a Ni-NTA column to remove the HRV3C protease.

**Crystallography**. The purified ENPP1-ΔSMB T238A protein (9 mg/ml) was mixed with 10 mM 2′3′-cGAMP (Invivogen) or 25 mM 3′3′-cGAMP (Invivogen) for 30 min on ice, and then crystallized at 20 °C by the hanging-drop vapor-diffusion method. The crystals with pApG were obtained under the crystallization conditions consisting of 17% PEG3350, 0.1 M Bis–Tris propane, pH 7.5, 0.2 M NaBr and 0.1 M MgCl₂. The co-crystals with 3′3′-cGAMP were obtained under the crystallization conditions consisting of 16% PEG3350, 0.1 M Bis–Tris propane, pH 8.5, 0.2 M NaBr and 0.1 M MgCl₂. Crystals were cryoprotected in the reservoir solution supplemented with 25% ethylene glycol and 25 mM 2′3′-cGAMP or 25 mM 3′3′-cGAMP. X-ray diffraction data were collected at 100 K on beamline BL41XU at SPring-8 (Hyogo, Japan), and were processed with XDS[30]. The structures were determined by molecular replacement with MOLREP[31], with the structure of mouse ENPP1 (PDB 4GTW) as the search model. Model building and refinement were performed with COOT[32] and PHENIX[33], respectively. Data collection and refinement statistics are summarized in Table 1.

**In vitro nucleotide-hydrolyzing assay**. The purified ENPP1 protein (200 nM) was incubated at 37 °C for 10 min with 2′3′-cGAMP or ATP (500 μM), in buffer containing 100 mM Tris-HCl, pH 9.0, 500 mM NaCl and 5 mM MgCl₂. The reaction mixture was heat-inactivated at 95 °C for 5 min, and then centrifuged at 86,400 × $g$ for 15 min. The supernatant was analyzed using a C-18 column (25 cm × 4.5 mm, 5 μM pore, Supelco Analytical) under two-step linear gradient conditions (0–10% Buffer B (2 column volumes), 10–50% Buffer B (2 column volumes); Buffer A: 0.1 M triethylammonium acetate/3% acetonitrile/97% H₂O, Buffer B: 45% methanol/45% acetonitrile/10% H₂O. The nucleotides were detected by the absorbance at 254 nm.

**Luciferase assay**. THP-1 cells from a human monocytic leukemia cell line were maintained in RPMI-1640 medium (Sigma), containing 10% FCS, 100 IU/ml penicillin G, and 100 μg/ml streptomycin, at 37 °C under a 5% CO₂ atmosphere. Transfection experiments were performed using GenomONE-GX (Ishihara Sangyo). THP-1 cells were co-transfected with the human ENPP1 expression plasmid, the pGL4-IFN-β-promoter-Luc plasmid, and the pRL-TK Renilla-Luc plasmid (Promega). At 24 h after transfection, the cells were lysed and the luciferase activity was measured with a GloMax 20/20 luminometer (Promega), using the Dual-Luciferase Reporter Assay System (Promega). The cell lysates were then immunoblotted with an anti-ENPP1 antibody (Cell Signaling, Cat. No. 2061, 1:1000 dilution) or an anti-tubulin antibody (Cedarlane, Cat. No.CLT9002, 1:3000 dilution). The uncropped images are shown in Supplementary Figure 3. All cell lines were negative for mycoplasma contamination.

## Data availability

The coordinates and structure factors have been deposited in the Protein Data Bank under the IDs 6AEK (pApG) and 6AEL (3′3′-cGAMP). Other data are available from the corresponding authors upon reasonable request.

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

## Acknowledgements

We thank the beamline staff at BL32XU and BL41XU of SPring-8, Japan, for assistance with data collection. We thank K. Ogomori and S. Okazaki for technical assistance. This work was supported by a grant from the Core Research for Evolutional Science and Technology Program, the Creation of Basic Chronic Inflammation, from the Japan Science and Technology Agency, to O.N.

## Author contributions

K.K. performed the purification, crystallization and biochemical analyses. G.K. assisted with the protein expression. K.K. and H.N. performed structural analyses, with assistance from S.H., H.H., and R.I. Luciferase assays were performed by D.O. and F.T.

The manuscript was written by K.K. and H.N., with help from all authors. K.K., H.N., and O.N. directed and supervised all of the research.

## Additional information

**Competing interests:** The authors declare no competing interests.

