## [Peer Review File · Nature Communications]

Reviewers' comments:

Reviewer #1 (Remarks to the Author):

The manuscript reports structural insights into the degradation of cGAMPs (2',3''-cGAMP and 3',3''-cGAMP) by mouse NPP1. The paper provides the hydrolysis mechanism of 2',3''-cGAMP based on structural and biochemical investigations. Furthermore, it explains why 2',3''-cGAMP, and not 3',3''-cGAMP, can be hydrolyzed by NPP1, analyzing binding poses of both cGAMPs. This paper provides a new perspective about the kinetic properties of NPP1, an enzyme, which is of great interest as a novel target for the immunotherapy of cancer and other potential indication.

Some points to be addressed:

1. The nomenclature has to be corrected throughout the manuscript to 2',3''-cGAMP and 3',3''-cGAMP
2. Title: cGAMP and ENPP1. Both are abbreviations. Please provide full names followed by their abbreviation in brackets.
3. Abstract: like Title, please give the full names of both cGAMP and ENPP1 and their abbreviations in brackets. Moreover, the abstract is not appropriate. It has to be rewritten providing more precise and clear information about the key results of the paper. NPP1 not only produces diphosphate (previous obsolete nomenclature: pyrophosphate) as a product, but another important product is AMP, which is further degraded by the enzyme CD73 to produce immunosuppressive adenosine. This has to be mentioned and discussed in the Introduction and in the Discussion (e.g. p. 7, second paragraph)
4. Page 3, line 32 and 33: Please give full names of TBK1, IRF3 and IFN- β followed by their abbreviations.
5. Page 3, line 34: isomers (not isoforms).
6. Page 3, line 41: Ecto-Nucleotide Pyrophosphatase Phosphodiesterase I \rightarrow Ecto- nucleotide pyrophosphatase / phosphodiesterase I
7. Page 3, line 46: pyrophosphate \rightarrow diphosphate (former nomenclature: pyrophosphate; current nomenclature: diphosphate)
8. Page 4, line 56: The crystallography was made with mouse ENPP1. Could you explain here the homology between human and mouse ENPP1 and give your opinion how your results can be transferred to human NPP1?
9. Page 5, line 91: Why was only the mutation of Ser514 investigated and not the mutation of His362 and Asn259? This would be interesting.
10. Page 5, line 94: Compare the binding orientation of 2',3''-cGAMP and ATP as an additional figure. This would explain the binding differences between both substrates.
11. Page 5, line 96: THP-1 cells \rightarrow human monocytic leukemia THP-1 cell line (abbreviation in brackets)
12. Page 7, line 133: Please specify that both AMP and GMP products are 5'-AMP and 5'-GMP, and not 2'-AMP and 3'-GMP.
13. Page 7, line 133: faintly \rightarrow temporarily
14. Page 7, line 138: pyrophosphate \rightarrow diphosphate
15. Page 7, line 147: cGAMP \rightarrow 2',3''-cGAMP
16. Figure 2: Correct nomenclature, also in other figures.
17. Supplementary figures: It would be nice, if they had also legends.
18. Supplementary Figure 1: As mentioned above, the numbering of hydroxy and phosphate groups has to be corrected.
19. Supplementary Figure 4: What is different between left and right Figures in A, B and C?
20. Has the crystal structure already been submitted to the PDB? Please mention the PDB ID of the submitted coordinates.
21. Is 3',3''-cGAMP acting as an enzyme inhibitor? Please check.
22. P.7, line 129: The flip-over was suggested already by ref. 23 – please mention and discuss this.
23. Compare the new crystal structure to the previously published model.

Reviewer #2 (Remarks to the Author):

The manuscript of NCOMMS-18-17881-T "Structural insight into cGAMP degradation by ENPP1" by Kazuki Kato, et. al. ... and Osamu Nureki has described high resolution crystal structures of ENPP1 (Ecto-Nucleotide Pyrophosphatase Phosphodiesterase 1) complexes with a few different substrates such as cGAMP of different linkage types bound to the active sites, the work is technically sound, and the results are convincing. The major finding of this structural work includes that ENPP1 could sequentially and specifically hydrolyze the two distinct 2'-5' and 3'-5' phosphodiester linkages in the 2'3'-cGAMP due to structural arrangement and zinc ion coordination with the 2'-5' linkage, and the complex structures showed nicely that although both 2'3'-cGAMP and 3'3'-cGAMP could bind almost identically in the active sites, but ENPP1 is not able to hydrolyze 3'3'-cGAMP due to the 3'-5' phosphodiester linkage in 3'3'-cGAMP has an unfavorable geometry for the in-line attack mechanism elucidated before, all of these providing insights into how ENPP1 could act as a bifunctional enzyme to degrade ATP for bone mineralization regulation and to hydrolyze the secondary messenger cGAMP for innate immune response.

The authors have extensively studied the mouse ENPP1 enzyme before and solved many types of NPP, ENPP and other relevant enzyme crystal structures, therefore they have accumulated lots of experience in this enzymes, which enabled them to expressed and purified these proteins in a right form to produce high resolution diffracting crystals. These two are basically the good and new results for this paper, it is a little thin for publication in NCOMMS, but it is OK for a brief communication paper.

There are some revisions the authors should do before further considerations:

1) There is a long standing controversy in the field that cGAS is a cytosolic enzyme to produce the cytosolic secondary messenger 2'3'-cGAMP, whereas ENPP1 is a type II transmembrane glycoprotein and located on the plasma membrane facing outside of the cell so that the ENPP1 probably will never reach the cytosolic 2'3'-cGAMP, thus the regulation modes needs to be address and further discussed, particularly in the light of ENPP1 knockout mice did not yield any immunity phenotype.

2) The following paper described a bacterial PDE (phosphodiesterase) specifically degrading 3'3'-cGAMP to produce pApG, it should be referred and discussed in the this manuscript.

Gao J, Tao J, Liang W, Zhao M, Du X, Cui S, Duan H, Kan B, Su X, Jiang Z. Identification and characterization of phosphodiesterases that specifically degrade 3'3'-cyclic GMP-AMP. *Cell Res.* 2015 May; 25(5): 539-50. doi: 10.1038/cr.2015.40.

Reviewer #1:

1. *The nomenclature has to be corrected throughout the manuscript to 2',3''-cGAMP and 3',3''-cGAMP.*

Thanks for this comment. Nevertheless, we would like to use 2'3'-cGAMP and 3'3'-cGAMP in the revised manuscript, since 2'3'-cGAMP and 3'3'-cGAMP are generally used for c[G(2',5')pA(3',5')p] and c[G(3',5')pA(3',5')p], respectively, in most of the literature, such as “Hydrolysis of 2'3'-cGAMP by ENPP1 and design of non-hydrolyzable analogs” Li *et al.*, *Nat. Chem. Biol.*, 2014.

2. *Title: cGAMP and ENPP1. Both are abbreviations. Please provide full names followed by their abbreviation in brackets.*

3. *Abstract: like Title, please give the full names of both cGAMP and ENPP1 and their abbreviations in brackets. Moreover, the abstract is not appropriate. It has to be rewritten providing more precise and clear information about the key results of the paper. NPP1 not only produces diphosphate (previous obsolete nomenclature: pyrophosphate) as a product, but another important product is AMP, which is further degraded by the enzyme CD73 to produce immunosuppressive adenosine. This has to be mentioned and discussed in the Introduction and in the Discussion (e.g. p. 7, second paragraph)*

4. *Page 3, line 32 and 33: Please give full names of TBK1, IRF3 and IFN- β followed by their abbreviations.*

Thank you for these helpful comments. According to the reviewer's suggestions, we have added the full names for these nomenclatures, and revised the abstract and introduction sections. We would like to use cGAMP and ENPP1 in the title, due to the character limit.

5. *Page 3, line 34: isomers (not isoforms).*

6. *Page 3, line 41: Ecto-Nucleotide Pyrophosphatase Phosphodiesterase I \rightarrow Ecto- nucleotide pyrophosphatase / phosphodiesterase I*

7. *Page 3, line 46: pyrophosphate \rightarrow diphosphate (former nomenclature: pyrophosphate; current nomenclature: diphosphate)*

We have modified these nomenclatures.

8. *Page 4, line 56: The crystallography was made with mouse ENPP1. Could you explain here the homology between human and mouse ENPP1 and give your opinion how your results can be transferred to human NPP1?*

The extracellular domains of the ENPP1 enzymes from mouse and human share ~80% sequence similarity, and the residues involved in cGAMP recognition are completely conserved, suggesting the conserved recognition mechanism of cGAMP. We have added the description about the similarity between the mouse and human ENPP1 enzymes in the discussion (Page 8).

9. *Page 5, line 91: Why was only the mutation of Ser514 investigated and not the mutation of His362 and Asn259? This would be interesting.*

In addition to cGAMP recognition, His362 is involved in coordination to a catalytic zinc ion, and the mutation of His352 abolished the phosphodiesterase activity of ENPP1 (Rik *et al.*, *JBC*, 2001). Asn259 recognizes the phosphate of nucleotide substrates, and the Asn230 mutation of mouse ENPP2 (equivalent to Asn259 of mouse ENPP1) abolished the phosphodiesterase activity of ENPP2

(Nishimasu *et al.*, *NSMB*, 2011). Therefore, we were unable to use the mutants of His362 and Asn259 to test the importance of the G-pocket for 2'3'-cGAMP recognition.

10. Page 5, line 94: Compare the binding orientation of 2',3''-cGAMP and ATP as an additional figure. This would explain the binding differences between both substrates.

According to the reviewer's suggestion, we have added a structural comparison between the cGAMP and AMP complexes in Supplementary Figure 2.

11. Page 5, line 96: THP-1 cells → human monocytic leukemia THP-1 cell line (abbreviation in brackets)

We have corrected it.

12. Page 7, line 133: Please specify that both AMP and GMP products are 5'-AMP and 5'-GMP, and not 2'-AMP and 3'-GMP.

We have specified AMP and GMP as 5'-AMP and 5'-GMP, respectively, in the text and the figures.

13. Page 7, line 133: faintly → temporarily

We would like to use "faintly" in this context.

14. Page 7, line 138: pyrophosphate → diphosphate

15. Page 7, line 147: cGAMP → 2',3''-cGAMP

We have corrected it.

16. Figure 2: Correct nomenclature, also in other figures.

As mentioned above, we would like to use 2'3'-cGAMP and 3'3'-cGAMP in the revised manuscript.

17. Supplementary figures: It would be nice, if they had also legends.

We have now combined the supplementary figures with the legends into one pdf file.

18. Supplementary Figure 1: As mentioned above, the numbering of hydroxy and phosphate groups has to be corrected.

As mentioned above, we would like to use 2'3'-cGAMP and 3'3'-cGAMP in our manuscript.

19. Supplementary Figure 4: What is different between left and right Figures in A, B and C?

As mentioned in the legend, Figures 2C, 2D, 2F and 4B-4D (the previous supplementary figures 4A-4D) are stereo views (wall-eyed), in which the left and right panels are rotated by -3 and 3 degrees relative to the y-axis, respectively.

20. Has the crystal structure already been submitted to the PDB? Please mention the PDB ID of the submitted coordinates.

We have deposited the structure coordinates in the PDB, under PDB IDs 6AEK (pApG) and 6AEL (3'3'-cGAMP), and added their PDB IDs in the revised manuscript (Page 15).

21. *Is 3,3'-cGAMP acting as an enzyme inhibitor? Please check.*

There are no reports showing that 3'3'-cGAMP acts as an inhibitor of ENPP1. Nonetheless, 3'3'-cGAMP could competitively inhibit the enzymatic activity of ENPP1 toward 2'3'-cGAMP, given that the adenosine moieties of 2'3'-cGAMP and 3'3'-cGAMP bind to the N-pocket in similar manners.

22. *P.7, line 129: The flip-over was suggested already by ref. 23 – please mention and discuss this.*

We added the description about the previously proposed model (Page 8).

23. *Compare the new crystal structure to the previously published model.*

We have added a structural comparison with the previous AMP complex in Supplementary figure 2.

Reviewer #2:

1) There is a long standing controversy in the field that cGAS is a cytosolic enzyme to produce the cytosolic secondary messenger 2'3'-cGAMP, whereas ENPP1 is a type II transmembrane glycoprotein and located on the plasma membrane facing outside of the cell so that the ENPP1 probably will never reach the cytosolic 2'3'-cGAMP, thus the regulation modes needs to be address and further discussed, particularly in the light of ENPP1 knockout mice did not yield any immunity phenotype.

As the reviewer pointed out, the catalytic domain of ENPP1 is exposed to the extracellular region, while cGAMP is a cytosolic second messenger. Therefore, some mechanisms should exist by which ENPP1 gains access to the cytosolic 2'3'-cGAMP, and further studies will be needed to elucidate how the extracellular enzyme ENPP1 degrades the cytosolic 2'3'-cGAMP at physiological levels. We added a statement about this point (Page 8) in the revised manuscript.

2) The following paper described a bacterial PDE (phosphodiesterase) specifically degrading 3'3'-cGAMP to produce pApG, it should be referred and discussed in the manuscript. Gao J, Tao J, Liang W, Zhao M, Du X, Cui S, Duan H, Kan B, Su X, Jiang Z. Identification and characterization of phosphodiesterases that specifically degrade 3'3'-cyclic GMP-AMP. Cell Res. 2015 May;25(5):539-50. doi: 10.1038/cr.2015.40.

We added a description about V-cGAPs in the discussion (Page 8).

REVIEWERS' COMMENTS:

Reviewer #2 (Remarks to the Author):

This revision of the manuscript is satisfactory to me, and I have no more questions and would like to recommend its publication.